# Transition-metal-free synthesis of pyrimidines from lignin β-O-4 segments via a one-pot multi-component reaction

Bo Zhang[1,6], Tenglong Guo[1,6], Zhewei Li [2,6], Fritz E. Kühn [3], Ming Lei [2], Zongbao K. Zhao [4], Jianliang Xiao[5], Jian Zhang[1], Dezhu Xu[1], Tao Zhang [1] & Changzhi Li [1✉]

Heteroatom-participated lignin depolymerization for heterocyclic aromatic compounds production is of great importance to expanding the product portfolio and meeting value-added biorefinery demand, but it is also particularly challenging. In this work, the synthesis of pyrimidines from lignin β-O-4 model compounds, the most abundant segment in lignin, mediated by NaOH through a one-pot multi-component cascade reaction is reported. Mechanism study suggests that the transformation starts by NaOH-induced deprotonation of Cα-H bond in β-O-4 model compounds, and involves highly coupled sequential cleavage of C-O bonds, alcohol dehydrogenation, aldol condensation, and dehydrogenative aromatization. This strategy features transition-metal free catalysis, a sustainable universal approach, no need of external oxidant/reductant, and an efficient one-pot process, thus providing an unprecedented opportunity for N-containing aromatic heterocyclic compounds synthesis from biorenewable feedstock. With this protocol, an important marine alkaloid meridianin derivative can be synthesized, emphasizing the application feasibility in pharmaceutical synthesis.

[1] CAS Key Laboratory of Science and Technology on Applied Catalysis, Dalian Institute of Chemical Physics, Chinese Academy of Sciences, Dalian 116023, China. [2] State Key Laboratory of Chemical Resource Engineering, Institute of Computational Chemistry, College of Chemistry, Beijing University of Chemical Technology, Beijing 100029, China. [3] Molecular Catalysis, Catalysis Research Center and Department of Chemistry, Technical University of Munich, Lichtenbergstr. 4, D-85748 Garching bei München, Germany. [4] Division of Biotechnology, Dalian Institute of Chemical Physics, Chinese Academy of Sciences, Dalian 116023, China. [5] Department of Chemistry, University of Liverpool, Liverpool L697ZD, UK. [6] These authors contributed equally: Bo Zhang, Tenglong Guo, Zhewei Li. ✉email: licz@dicp.ac.cn

The depletion of fossil resources and rising environmental concerns have led to great interest in biomass utilization[1,2]. Lignin, one of the three major components of lignocellulose, has received unique attention because it is regarded as a promising renewable source for aromatic chemicals[3–6]. So far, numerous efforts have been dedicated to controllable cleavage of the C-O and C-C bonds in lignin to obtain low molecular weight aromatics through introducing hydrogen or oxygen[1,7,8]. More recently, introducing heteroatom such as nitrogen during lignin depolymerization leading to heteroatom-containing aromatics received much attention, as it is of great potential to expand the product portfolio and to improve the economics of lignin conversion. Importantly, N-heterocyclic aromatic compounds represent a class of vital precursors to pharmaceuticals, dyes and hydrogen storage materials[9–11]; therefore, production of lignin-derived N-containing aromatics may offer sustainable routes for those value-added compounds.

State-of-the-art N-participated lignin conversion is limited to the production of N-containing chemicals from lignin-derived monomers or modified dimer model compounds (Fig. 1)[12–22]. For instance, hydrogenolysis or oxidation of lignin β-O-4 model compounds catalyzed by transition metals affords monophenols for subsequent amination processes using organic or inorganic N-sources via one or multiple steps to produce different N-containing compounds (Routes 1–3 in Fig. 1). In another case, oxidative modification of lignin β-O-4 dimers to ketone derivatives in the presence of oxidant agents such as DDQ (2,3-dichloro-5,6-dicyano-1,4-benzoquinone)[23] and TEMPO (2,2,6,6-tetramethyl-1-piperidyloxy)[24], followed by different amination processes has also been reported to generate N-containing aromatics (Routes 4–6 in Fig. 1). In brief, routes developed so far for the conversion of β-O-4 model compounds, the most abundant segments in lignin, involve hydrogenolysis or oxidative pretreatment. External oxidant or reductant species are essential in such multi-step processes. Moreover, most products are mononitrogen-containing aromatics. Our recent progress achieves the synthesis of benzylamines from β-O-4 model compounds, and the feasibility of the production of benzylamines from lignin has also been demonstrated by a two-step process[25]. To the best of our knowledge, no literature reports the direct conversion of β-O-4 model compounds to aromatic heterocyclic compounds containing multiple nitrogen atoms in the absence of transition metal catalysts and external redox reagents, due to the extremely complicated reaction path, and the incompatible catalysis for C-O bond cleavage, C-N formation, and aromatic nitrogen-heterocyclic ring construction.

Pyrimidines are one type of such N-containing heterocyclic compounds that exhibit broad biological (such as antibacterial, antiallergic, anti-HIV, and antitumor) activities[26], and have been widely used to design new physiologically and pharmacologically active compounds[27]. Hence, assembling a pyrimidine core has driven considerable investigations. Extensive studies have focused

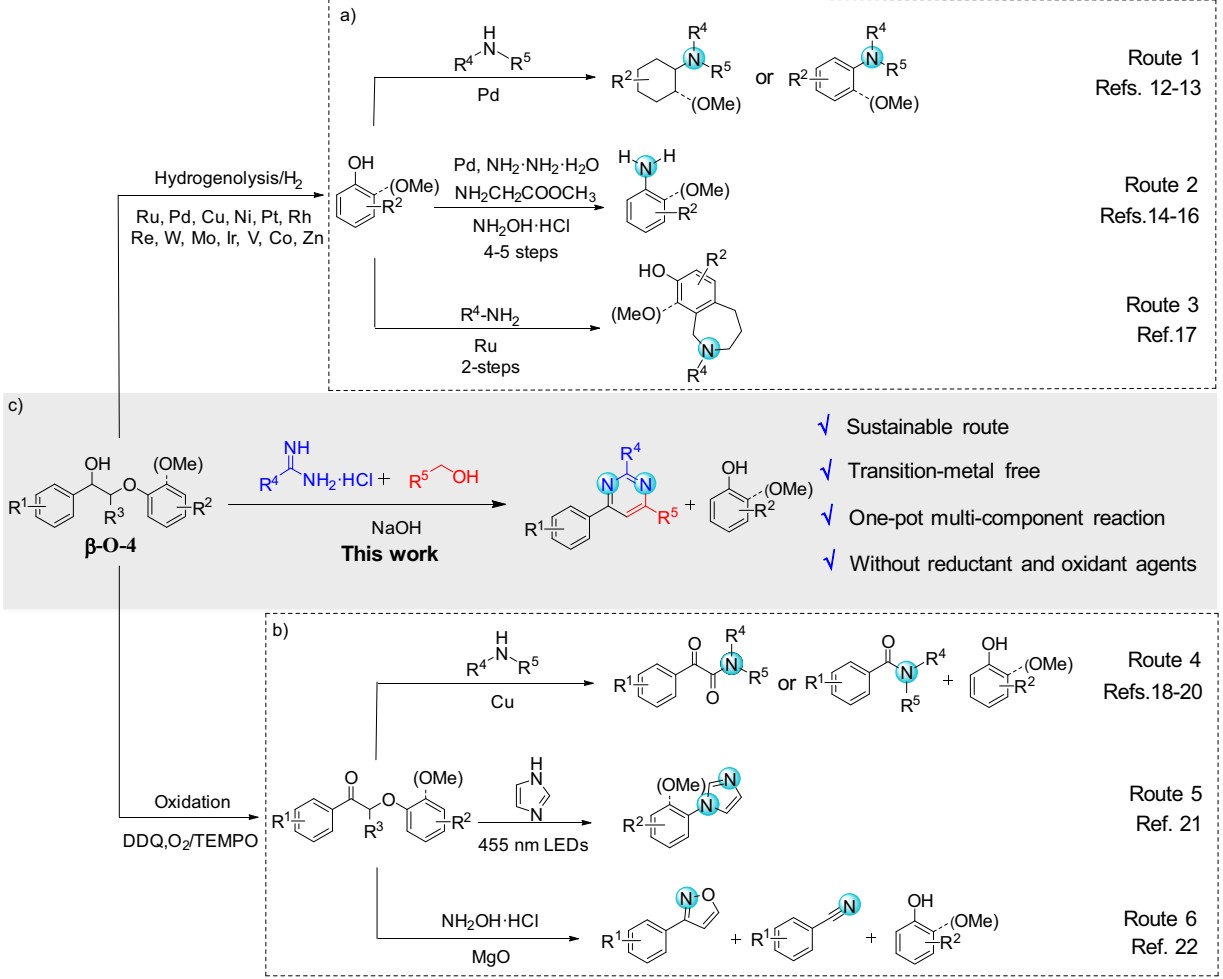

**Fig. 1 Production of N-containing aromatics from lignin β-O-4 model compounds. a** Amination of monophenols after hydrogenolysis of phenolic β-O-4 model compounds; **b** conversion of modified β-O-4 model compound to produce N-containing aromatics; **c** direct conversion of β-O-4 model compound without transition-metals to produce pyrimidines.

on amidine reactions with coupling partners such as 1,3-dicar-bonyl derivatives, α,β-unsaturated ketones, alkynones, 1,2,3-triazines[28], and alcohols in the presence of organometallic catalysts based on Ir[29], Ru[30], Re[31], Mn[32,33], and Ni[34]. However, most of the methods described so far suffer from employing transition metal catalysts and additives, with complicated ligands, and using non-renewable substrates. On account of the functionalized aromatic scaffold of pyrimidines in structure, the abundant renewable lignin resources would serve as excellent, carbon-neutral starting materials for the development of sustainable routes to pyrimidines, which would also be promising to achieve the green chemistry demand to the environment.

In this work, based on the above-listed points, the development of a robust methodology for one-pot pyrimidines synthesis from lignin β-O-4 model compounds is described (Fig. 1). The described route is mediated by NaOH in the absence of either transition-metal catalysts or external oxidants/reductants. It combines highly coupled multi-step reactions, including selective C-O bond breaking, aldol condensation, C-C/C-N bond formation, and dehydrogenative aromatization. Thus, it provides an opportunity to prepare pyrimidines from renewable feedstock, which may promote further efforts to explore the abundant lignin resources for the construction of N-containing aromatic heterocyclic compounds.

## Results

**Development of the reaction.** Lignin β-O-4 linkages represent a predominant portion of all linkages between the primary units. Successful breaking β-O-4 units in model compounds should offer guidance for the depolymerization of realistic lignin. Thus, a typical lignin β-O-4 model compound **1a**, accompanied by ben-zamidine hydrochloride (**2a**) and benzyl alcohol (**3a**) has been initially employed to synthesize 2,4,6-triphenylpyrimidine **4a** (Fig. 2, entry 1, and Supplementary Table 1). We found that the presence of a base played a crucial role in this reaction. In the absence of base, the reaction did not occur (Supplementary Table 1, entry 1). While NaOH was found to be the most efficient base in stark contrast to KOH, sodium tert-butoxide (t-BuOK), $Cs_2CO_3$, and $CH_3CH_2ONa$ (Supplementary Table 1, entries 2–6). After screening the reaction parameters including solvents, base loading, reaction temperature, and time (Supplementary Tables 1–3 and Supplementary Fig. 3), the optimized condition was identified, leading to 95% gas chromatograph (GC) yield (93% isolated yield) of **4a** based on the amount of **2a**, along with 99% GC yield (95% isolated yield) of guaiacol **5a** (Fig. 2, entry 1). In this case, 40.4 wt% of **1a** is incorporated in **4a** formation (for a detailed calculation process, see Supplementary Fig. 2). After the reaction, the solvent tert-amyl alcohol can be easily distilled from the reaction mixture for recycling because the boiling point of tert-amyl alcohol (101.8 °C) is much lower than that of all the reactants (e.g., **1a**: 398 °C; **2a**: 208 °C; **3a**: 205 °C) and the products (**4a**: 330 °C; **5a**: 205 °C). Moreover, the reaction can be proceeded at a higher substrate concentration (Supplementary Table 1, entry 16) but the yields of the targeted products decreased due to severer side reactions. Hence, our strategy not only provides access to a pyrimidine product in high isolated yield, but also achieves co-production of isolated guaiacol in excellent yield, which markedly increases the atom economics.

To examine the generality of this protocol, the activity of various lignin β-O-4 model substrates was explored. Substrates with different functional groups on the aryl ring (Fig. 2) are found as fragments in different lignins[1]. Under optimized condition, all β-O-4 model compounds were completely consumed to afford the corresponding pyrimidines and monophenols. **2a** and **3a** reacting with β-O-4 model compounds **1a**–**1f** bearing methoxy groups on both aryl rings afforded moderate to excellent isolated

yields of pyrimidine products **4a** (74–93%) or **4b** (64–90%) along with phenol derivatives **5a**–**5c** (68–99%) (Fig. 2, entries 1–6), indicating that the formation of pyrimidine heterocycles occurred associated with selective C-O bond cleavage and C-C/C-N bonds construction in a one-pot fashion. One methoxyl group on the O-terminal aryl ring exhibited a positive influence on the reaction efficiency (90–93%) (Fig. 2, entries 1 and 4) compared to that bearing no functional group (64–75%) (Fig. 2, entries 2 and 5) and 2,5-dimethoxy substituents (74%) (Fig. 2, entries 3 and 6), while methoxyl substitution on the C-terminus aryl ring had little impact on the product yield. Specifically, the highly substituted β-O-4 model compound **1g** containing γ-OH functionality was also tolerated, smoothly providing the target product **4c** in 38% yield (Fig. 2, entry 7) despite it has a more complicated structure and contains higher steric hindrance compared to **1a**–**1f**. Based on the above-described results, it can be concluded that this protocol leads to the successful cleavage of various β-O-4 model compounds and to the synthesis of pyrimidines with good to excellent yields in a one-pot fashion, which thus provides an opportunity for the utilization of lignin to produce value-added N-heterocyclic compounds.

This reaction system has proven to be effective not only for a variety of amidine hydrochlorides, but also for a broad range of primary alcohols. As shown in Fig. 3, eight amidine hydrochlorides **2** have been successfully employed in the transformation, and the yields of corresponding pyrimidine products (**4d**–**4h**, 80–94%) from aryl amidine hydrochlorides (Fig. 3, entries 1–5) were relatively higher than that from aliphatic amidine hydrochloride (Fig. 3, entry 7, **4j** yield 64%). The low yield of pyrimidine **4j** is probably due to a negative electronic effect of the ethylamine backbone. Particularly, guanidine hydrochloride also exhibited a high reactivity under standard conditions, giving **4i** in 83% yield (Fig. 3, entry 6). In addition, entries 1–5 in Fig. 3 show that the substituents on the aryl rings of aryl amidine hydrochlorides do not obviously affect the reaction efficiency, regardless of the presence of electron-donating substituents (-$CH_3$, -$OCH_3$) or electron-withdrawing substituents (-Cl, -F). A broad range of primary alcohols **3** has also been tested (entries 8–12 in Fig. 3). All the reactions proceeded fairly well and yielded 71–92% of pyrimidines **4k**–**4o**. Notably, two heteroaryl primary alcohols underwent the same reactions to generate high yields of **4n** (80%) and **4o** (71%), respectively (entries 11–12). Therefore, the above results suggest that such an original protocol for bio-pyrimidines synthesis from lignin β-O-4 segment shows great versatility for all three reaction components. Various lignin model compounds, amidine hydrochlorides, and primary alcohols containing different functional groups are effective in the transformation, representing an interesting breakthrough for the functionalization of lignin products.

To further determine the compatibility of the reaction system, a β-O-4 polymer mimicking natural lignin was prepared and was used as a substrate for the synthesis of pyrimidine derivative. Though the one-pot, direct conversion of the β-O-4 polymer to pyrimidine was not achieved by reason that the reactive OH group at the para-position of β-O-4 polymer easily reacted with NaOH to produce the sodium phenolate salt under the base environment, an alternative three-step process composed of our key reaction was developed, which achieved 66% overall yield of pyrimidine derivative **4p** based on β-O-4 polymer (Supplementary Fig. 1). First, binuclear rhodium complex-catalyzed mild depolymerization of β-O-4 polymer led to 4-hydroxyacetophenone (compound **I**) in 75% isolated yield, then 4-hydroxyacetophenone reacted with benzyl bromide in the presence of $K_2CO_3$ to afford 1-(4-(benzyloxy)phenyl)ethan-1-one (compound **II**) in 95% isolated yield, which subsequently reacted with benzamidine hydrochloride and benzyl alcohol, successfully providing pyrimidine derivative **4p** in 92% isolated yield (66 wt% yield based on polymer).

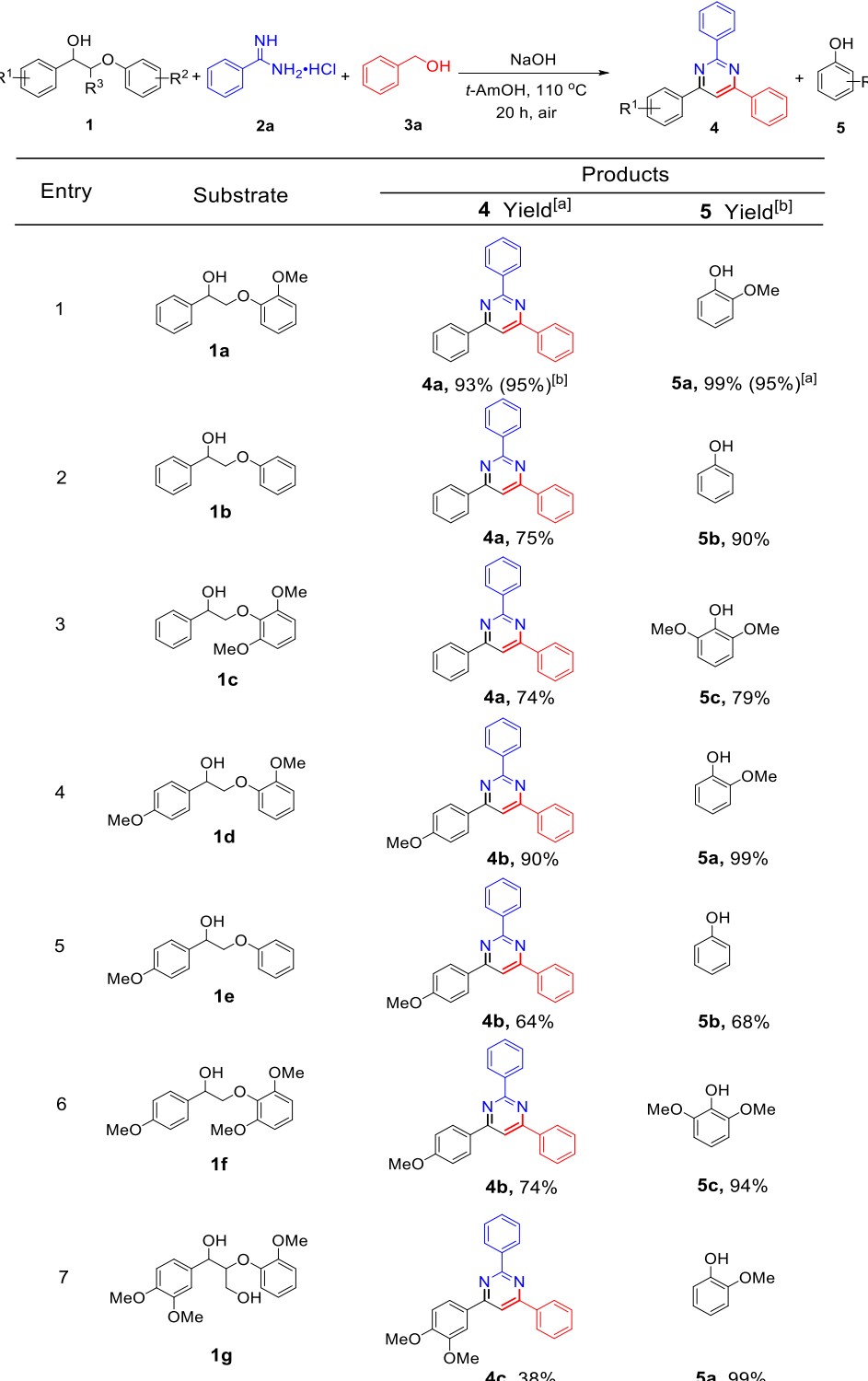

**Fig. 2 Conversion of different lignin β-O-4 model compounds to pyrimidines.** Conditions: **1** (0.4 mmol), **2a** (0.2 mmol), **3a** (0.4 mmol), NaOH (1.6 mmol), and *tert*-amyl alcohol (abbreviation: *t*-AmOH, 4.0 mL) were mixed in the air at 110 °C, reaction time (*t*) = 20 h; unless otherwise specified, the yields of **4** and **5** were calculated based on the amount of **2a** and **1**, respectively. [a]Isolated molar yield; [b]GC molar yield was determined by GC-FID using mesitylene as an internal standard.

**Mechanistic studies**. To gain insight into the reaction mechanism, several control experiments were performed to identify possible reaction intermediates. First, treatment of **1a** alone under otherwise identical conditions yielded 71% of acetophenone **6** and 82% of guaiacol **5a** within 1 h (reaction 1 in Fig. 4a), indicating that base-catalyzed C-O cleavage of **1a** might be the initial reaction step[35]. This assumption was confirmed by the three-component reaction using compounds **6**, **2a**, and **3a** as substrates (reaction 2 in Fig. 4a), which afforded a similar yield of **4a** (99%) to that in Fig. 2, entry 1. Hence, in the whole reaction process,

**Fig. 3 Scope of amidine hydrochlorides and primary alcohols.** Conditions: **1a** (0.4 mmol), **2** (0.2 mmol), **3** (0.4 mmol), NaOH (1.6 mmol), *tert*-amyl alcohol (abbreviation: *t*-AmOH, 4.0 mL) were mixed in the air at 110 °C, *t* = 20 h; the yields of **4** and **5a** were calculated based on the amount of **2** and **1a**, respectively. [a]Isolated molar yield; [b]GC molar yield was determined by GC-FID using mesitylene as an internal standard.

NaOH facilitates the cleavage of β-O-4 model compounds to release **6** as an important intermediate for subsequent reactions. Moreover, when benzyl alcohol **3a** (Fig. 2, entry 1) was replaced by benzaldehyde **7** as a substrate, the respective reaction (reaction 3 in Fig. 4a) also afforded **4a** in 82% yield, suggesting that dehydrogenation of **3a** into the corresponding aldehyde **7** is another prerequisite step under base condition for the transformation[36]. An additional experiment using **6**, **2a**, and **7** as the substrates (reaction 4 in Fig. 4a) gives a direct evidence by showing that the desired product **4a** is obtained in a high yield of 99%. It is well known that a cross-aldol condensation readily occurs between aldehyde and ketone under basic conditions[37]. We therefore assumed that aldol condensation between **6** and **7** would occur to generate chalcone (compound **8** in Fig. 4a) as an intermediate, which would further react with **2a** to form the target product 2,4,6-triphenylpyrimidine **4a**. This assumption was supported by another reaction using **8** and **2a** as the substrates, which afforded **4a** in 59% yield (reaction 5 in Fig. 4a). On the basis of the above results, a tentative multi-step consecutive

pathway can be proposed: the transformation starts with cleavage of the C-O bond in the lignin β-O-4 model compound **1a** to release **6**, accompanied by the dehydrogenation of **3a** to **7**. Then **6** and **7** undergo a cross-aldol condensation reaction to yield intermediate **8**, which subsequently reacts with **2a** to form the six-membered ring intermediate **9** via cyclization reaction. Finally, **9** undergoes an intramolecular dehydrogenative aromatization to furnish **4a** (Fig. 4b). It is worth noting that NaOH plays an important role in each step, *viz*. selective C-O bond cleavage, cross-aldol condensation, dehydrogenation, and dehydrogenative aromatization, of the whole transformation, and no other catalyst is required, as further supported by the DFT calculation described below.

To obtain insight into the role of NaOH and further disclose the underlying mechanism, DFT calculations have been carried out. Considering that polar solvent *tert*-amyl alcohol (*t*-AmOH) was used and sodium ions exist in the reaction solution, a reaction mediated by sodium hydroxide instead of Zundel anion[38] or hydroxide ion[39] was proposed during DFT calculations (Fig. 5).

**Fig. 4 Mechanistic studies of pyrimidine formation from lignin β-O-4 model compound. a** Control experiments and **b** the proposed pathway.

In fact, sodium hydroxide can either deprotonate the $C_\alpha$-H bond along path A or the O-H bond of $C_\alpha$H-OH moiety along path B to form the corresponding intermediates **A2** and **B2**, respectively, starting from β-O-4 model compound **1a** (Fig. 5a). For $C_\alpha$-H bond cleavage, **A2** produces **5a** and **A4** via a proton transfer step; in the other possible pathway adopting O-H bond cleavage, **B2** proceeds a cascade reaction of epoxidation, ring cleavage, and dehydration to form **A4** (**B2**→**B3**→**5a**+**B5**→**B7**→**A4**), which

further tautomerizes into acetophenone **6** catalyzed by base. These calculation results indicate that the $C_\alpha$-H bond and $C_\beta$-O bond would be broken simultaneously via the transition state (**TS**) **TSA1-2** activated by sodium hydroxide (Fig. 5b, path A). The free energy barrier for this C-H/C-O activation step is calculated to be 25.2 kcal/mol. The sodium ion could polarize the β-O atom to lead to a $C_\beta$-O bond cleavage (paths E and F in Supplementary Fig. 22), further illustrating that a sodium ion plays a crucial role in this

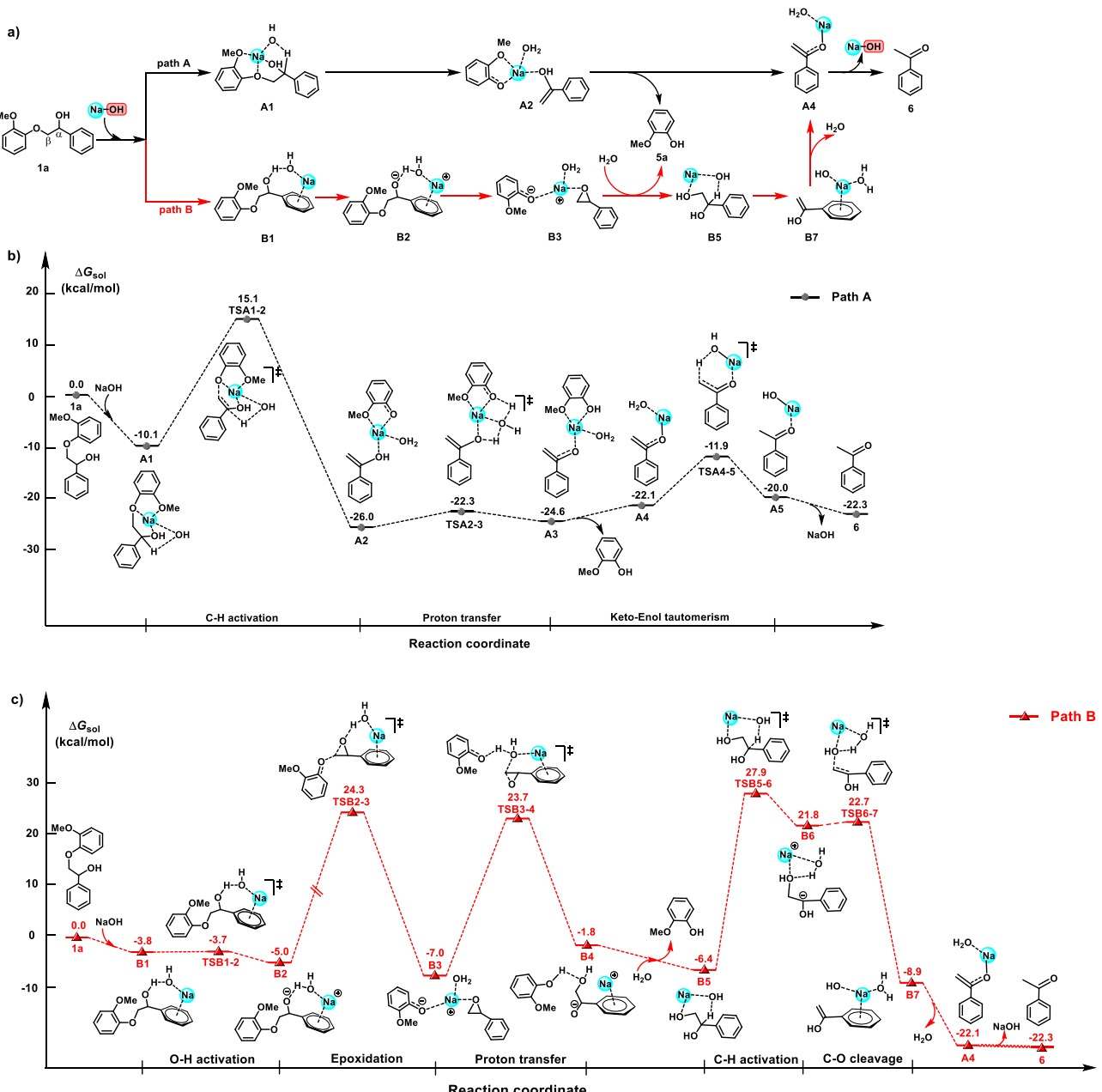

**Fig. 5 Computational analysis of the C-O bond cleavage of compound 1a. a** Proposed two pathways. **b** Computational analysis of path A (unit: kcal/mol). **c** Computational analysis of path B (unit: kcal/mol).

reaction. Then the proton of the enol moiety of **A2** is transferred to the phenol moiety via proton transfer, with a free energy barrier of 3.7 kcal/mol. **A3** releases phenol to produce **A4**. The proton from the water moiety of **A4** is transferred to the enol ion again via **TSA4-5** to complete the keto-enol tautomerism with a free energy barrier of 10.2 kcal/mol. Apparently, intermediate **6** can be easily generated along path A, and the C-H activation step is the rate-determining step in the process from β-O-4 model compound **1a** to **6**. Compared to path A, the calculated energy barrier of the epoxidation step from **B2** to **B3** in path B is 29.3 kcal/mol (Fig. 5c, path B), and that for the C-H activation step from **B5** to enol **A4** is 34.3 kcal/mol. Thus, path A is much more favorable for the $C_\beta$-O bond cleavage of **1a** (**A1→A2→5a+A4**) than path B (**B1→B2→B3→5a+B5→B7→A4**).

The success of this one-pot, multi-step reaction towards pyrimidines synthesis is governed by two dehydrogenation steps

(Figs. 6 and 7): benzyl alcohol **3a** dehydrogenation to benzalde-hyde **7** and dehydrogenative aromatization of **9** to **4a**.

Because the above dehydrogenations are base promoted[40], it was unclear whether this process employs a hydrogen acceptor or not. Therefore, a control experiment under argon atmosphere was conducted, which resulted in a much lower **4a** yield (68%, reaction 6 in Fig. 4a) compared to that in the air (99%, reaction 2 in Fig. 4a), illustrating that oxygen in air acted as a hydrogen acceptor in the reaction. It also should be noted that 1-phenylethanol was detected, indicating that intermediate **6** (acetophenone) acted as another hydrogen acceptor. With the aim to verify this speculation and better understand the dehydrogenation mechanism, further DFT calculation was performed. The results, summarized in Fig. 6, show that benzyl alcohol **3a** reacts with sodium hydroxide via **TSC1-2** to generate **C2** with an energy barrier of 0.2 kcal/mol. In path C using $O_2$ as the hydrogen acceptor, **C2** interacts with $O_2$ to achieve **C3**,

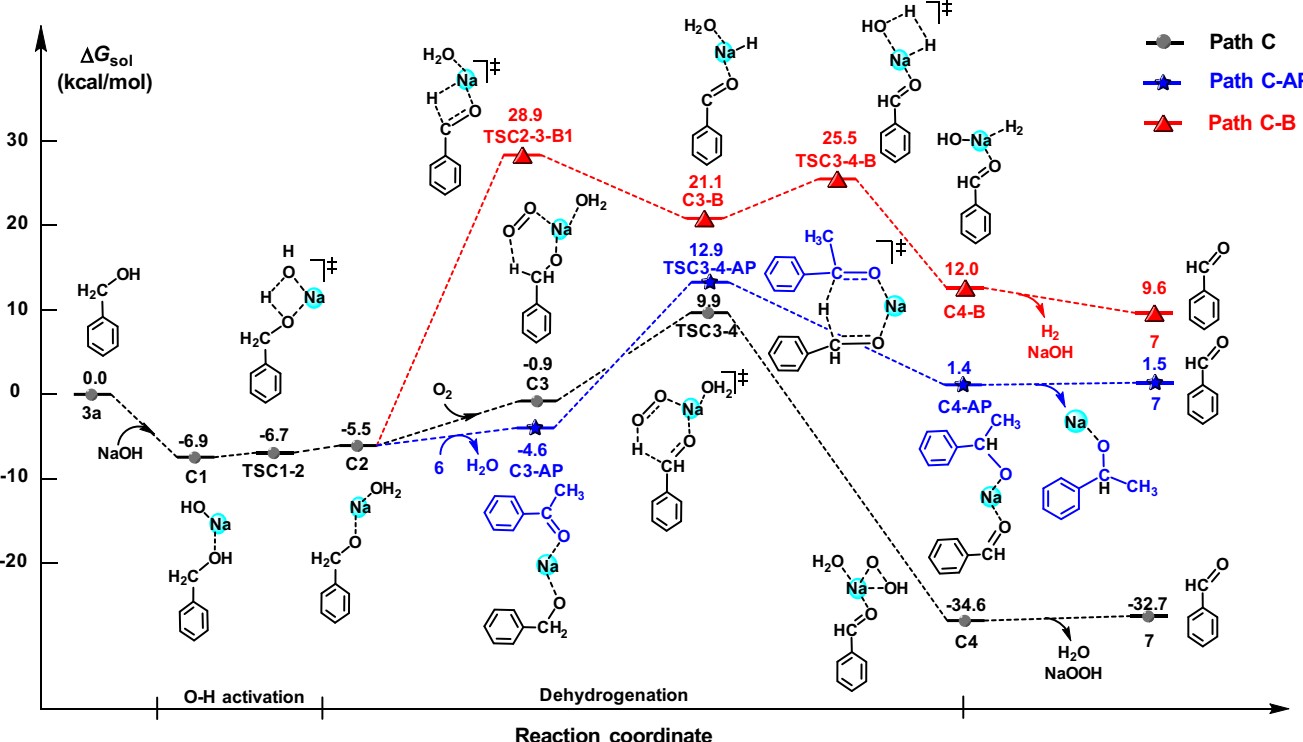

**Fig. 6 The Gibbs free energy profiles for the dehydrogenation of benzyl alcohol 3a.** Using O₂ in the air as hydrogen acceptor (path C), acetophenone **6** as hydrogen acceptor (path C-AP), and via hydrogen-acceptor-free pathway (path C-B) (unit: kcal/mol).

followed by the dehydrogenation of **C3**. The energy barrier for the dehydrogenation step via **TSC3-4** is 10.8 kcal/mol. And in path **C-AP** using acetophenone **6** as a hydrogen acceptor, the energy barrier of the dehydrogenation via **TSC3-4-AP** is 17.5 kcal/mol. This is consistent with the experimental observation in reaction 2 Fig. 4a. In addition, the acceptor-free dehydrogenation pathway was also calculated under the same condition, giving two possibilities. As shown in Fig. 6 and Supplementary Fig. 23, for acceptor-free dehydrogenation, the energy barrier via **TSC2-3-B1** would be 34.4 kcal/mol, which is higher than in path **C**. Moreover, the experiment results further suggest that no hydrogen was detected during the reaction. Combining the DFT calculation and experimental results, it can be concluded that acceptor-free dehydrogenation did not occur for the dehydrogenation of benzyl alcohol.

Similarly, the dehydrogenation of intermediate **9** using both O₂ in the air (path D) and **6** (path D-AP) as hydrogen acceptors was also studied. For comparison, an acceptor-free dehydrogenation pathway (path D-B) was also investigated by DFT calculation. As shown in Fig. 7, **9** first reacts with sodium hydroxide via **TSD1-2** to generate **D2**. Then **D2** follows path **D-B** to generate pyrimidine **4a** in the absence of a hydrogen acceptor with an energy barrier of 30.1 kcal/mol. Alternatively, **D2** goes through **D3** and **D4** to form **4a** in the presence of air with a much lower energy barrier of 3.2 kcal/mol, which is lower than that of path **D-AP** from **D3-AP** and **D4-AP** (13.0 kcal/mol), and H₂ is not detected during the reaction. Apparently, the pathway from **9** to **4a** is also not an acceptor-free dehydrogenation reaction. The DFT results summarized in Figs. 6 and 7 illustrate that O₂ in the air acts as a hydrogen acceptor for the dehydrogenation of benzyl alcohol **3a** to **7** and **9** to **4a**. In addition, the by-product NaOOH in path C and path D could oxidize **3a** or **9** to release two NaOH molecules (Supplementary Fig. 24).

**Application in synthesis of meridianin derivatives**. The potential application of this protocol is further highlighted by the preparation of pharmaceutical intermediates, namely meridianin

derivatives. Meridianin derivatives are an important class of natural marine alkaloids that display unique bioactivities, such as high antitumor activity, and therefore are widely used in the pharmaceutical industry[41,42]. Typically, meridianin derivatives are synthesized either by a multi-step condensation of substituted indoles with guanidines[42–44], or by a Suzuki coupling reaction between indolyl boronates and halopyrimidines over palladium catalysts[45]. Based on the above-described route for bio-based pyrimidines, here we established an interesting protocol that allows meridianin derivative production, starting from a lignin β-O-4 model compound through a two-step process (Fig. 8). First, intermediate **10** was produced in 69% yield upon isolation by treatment of β-O-4 model compound **1d**, guanidine hydrochloride **2g** with (1-benzyl-1H-indol-3-yl) methanol **3g**. Subsequently, debenzylation of **10** with *t*-BuOK/DMSO under an oxygen atmosphere successfully afforded the meridianin analog **11** in 70% yield (21.5 wt% based on lignin model compound **1d**, Supplementary Fig. 2). Such a simple methodology does not require any transition-metal catalyst, thus providing a cost-effective alternative to synthesize value-added meridianin derivatives.

## Discussion

In summary, we have shown an efficient synthesis of functionalized pyrimidines through multi-component reaction of lignin β-O-4 model compounds with amidine hydrochlorides and primary alcohols under transition-metal-free conditions. A highly coupled cascade process, including cleavage of C-O bonds, alcohol dehydrogenation, aldol condensation, and dehydrogenative aromatization reaction has been established to selectively afford a wide range of alkylated and arylated heterocyclic pyrimidines in a one-pot fashion. The methodology can also be applied to assemble meridianin derivatives, which underlines the applicability of this protocol for the synthesis of pharmaceuticals. This protocol paves the way toward applications of lignin, creating a bridge between renewable biomass and pharmaceutical synthesis, and pushes

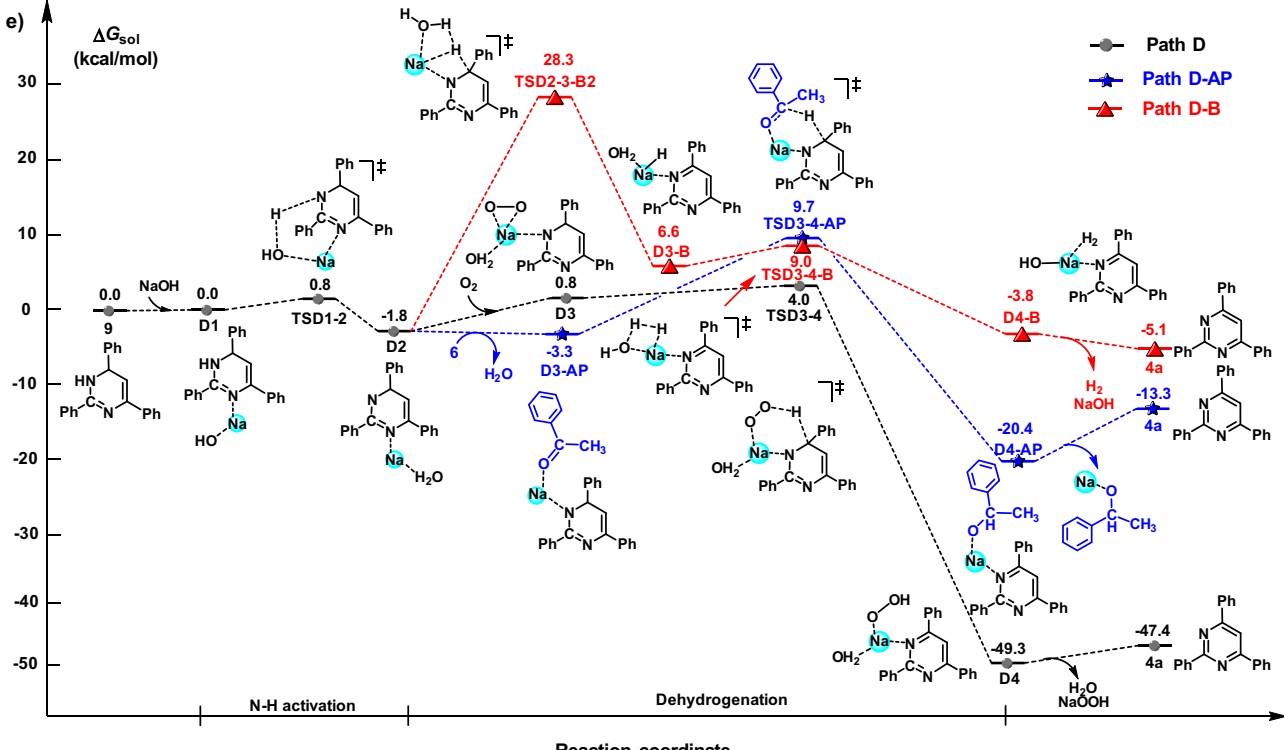

**Fig. 7 The Gibbs free energy profiles for the dehydrogenation of intermediate 9.** Using $O_2$ in the air as a hydrogen acceptor (path D), acetophenone **6** as a hydrogen acceptor (path D-AP), and via hydrogen-acceptor-free pathway (path D-B) (unit: kcal/mol).

**Fig. 8 Synthesis of a meridianin derivative from lignin β-O-4 model compound.** Step one:synthesis of compound **10** from a mixture **1d**, **2g** and **3g** by this protocol; step two: debenzylation of **10** in the presence of t-BuOK/DMSO under an oxygen atmosphere to afford meridianin derivative **11**.

forward one-pot conversion of lignin to value-added pharmaceutical molecules. Further exploration on native lignin is undergoing.

## Methods

**Typical procedure for the pyrimidine synthesis from lignin β-O-4 model compound.** Lignin model compound (0.4 mmol), the primary alcohol (0.4 mmol), benzamidine hydrochloride (0.2 mmol), NaOH (1.6 mmol), internal standard mesitylene (8 mg), and t-AmOH (4 mL) were placed in the pressure tube (35 mL). The mixture was sealed and heated to 110 °C for 20 h. After the reaction, the solution was cooled to room temperature, and ethyl acetate (6 mL) was added to the mixture. Then hydrochloric acid (2 M) was used to acidify the aqueous solution to pH = 1. The organic phase was analyzed by GC-FID to determine the yield of phenol derivatives. Then the solvent was evaporated under reduced pressure, and the crude products were purified by column chromatography using petroleum ether/ethyl acetate (9:1) to obtain the desired products.

**Details of DFT calculations.** In this work, ωB97X-D[46] functional was used for the DFT calculation, which considered the dispersion correction. And the Pople-type triple-ζ split-valence basis sets 6–311+G(d,p) are used for the optimization of all structures. The solvation effect of *tert*-amyl alcohol ($\varepsilon = 5.78$) was simulated by the SMD[47] continuum solvent mode. The Cartesian coordinates of all optimized structures are given in the Supporting Information. The calculations were performed using Gaussian 09 program[48]. All transition states were confirmed with only one imaginary frequency integrated with intrinsic reaction coordinates[49] calculations. All energies discussed above are Gibbs free energies calculated at 298.15 K. For more information about the DFT calculations, see Supplementary information 1.5 and Supplementary Data 1–3.

## Data availability

The data that support the findings of this study are available within the article, the Supplementary information, and Supplementary Data 1–3. Any other relevant data are also available from the authors upon reasonable request.

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

## Acknowledgements

Support from National Key R&D Program of China (2019YFC1905300), the National Natural Science Foundation of China (22078317, 22108272, 22073005, 21721004, 21690083), the Strategic Priority Research Program of the Chinese Academy of Sciences (XDB17020100), and the 2017 Royal Society International Collaboration Award (IC170044) is gratefully acknowledged.

## Author contributions

C.L. and T.Z. conceived the study and directed the project; B.Z. and T.G. designed and performed the experiments; Z.L. and M.L. performed the DFT calculation. B.Z., C.L., and T.Z. wrote the manuscript. F.E.K., Z.K.Z., J.Z., D.X., and J.X. improved the manuscript. All the authors discussed the results and commented on the manuscript.

## Competing interests

The authors declare no competing interests.
