## [Peer Review File · Nature Communications]

REVIEWER COMMENTS

Reviewer #1 (Remarks to the Author):

In the paper "Transition-metal-free synthesis of pyrimidines from lignin β -O-4 segments via a one-pot multi-component reaction" an approach to address the sourcing, synthesis, and the mechanism behind the formation of pyrimidines containing bio-based fine chemicals is presented. This paper is important for bridging the gap between the fundamental problems of fossils dependence and implementon of bio-based alternatives into the market. Starting from model compounds mimicking lignin main unit, the authors performed elegant cascade of reactions to generate pyrimidine derivatives in fair yields (based on lignin part) and reasonable selectivity. The mechanism of the reaction is partially studied experimentally and thoroughly supported by the computational analysis. The synthesis of a meridianin derivative performed to highlight applicability of the method.

This is a challenging topic and it is well addressed by the authors.

I have divided my comments into major A# and minor B#:

A1. The yield of all compounds is calculated based on a benzamidine derivatives. It would be useful to point out what is the wt% incorporation of the lignin derived part in the final products since both they are used in excess and stated conversion is 100%. The authors should include that information in the main text and also for the best presented case and meridianin derivative. For example, for the derivative 4c, that is the most relevant one, the yield would be 19% (ca 14.7 wt% of the initial lignin model is incorporated).

A2. Among model compounds there are no examples of the terminal β -O-4 groups (non-methylated hydroxy group in the para-position to the main chain/backbond). That is an important part that should be added and such model compound need to be tested. Terminal and internal groups are known to react in a very different way.

A3. It is stated in the article that biomass derived lignin is foreseen. Therefore, a model β -O-4 polymer need to be tested. That would demonstrate the solvent system and reactions conditions are compatible with high molecular weight compounds.

A4. There is no supplementary information (procedure, etc.) for the Reactions presented in Figure 2 in the SI. Please add it.

A5. There is no supplementary information for the synthesis presented in Figure 4. Please add it (amounts, purification etc.)

A6. How the amount of compound 3a was justified? Is there any optimization available?

A7. Authors proposed oxygen as the oxidizing agent in the reaction supported by calculations. Please perform reactions Table 1, entry 1 and Figure 2, line a) under inert atmosphere. That will both strength the statement and also exclude the possibility that fragments formed from 1a are possible oxidants.

B1. Page 6. Part of the text is devoted to the justification of the guaiacol formation and it usefulness as a chemical. I would recommend to remove that part since describe process won't result in this product when a relevant starting material will be used.

B2. The part where the mechanism is studied is lacking of references. Most of the reactions are

known and there is no literature cited. Please check for all the steps discussed in this part.

B3. Table S1. Please remove duplicates.

B4. In the descriptions of the spectral data in the SI there is no references to the known compounds spectral data. Was the data compared and verified that it is matching the literature? Please add references and verified that obtained spectral data is in accordance with the previously reported.

Overall, the authors have presented a novel system that potentially can convert lignin from wood to fine chemicals containing pyrimidines. There is considerable novelty and value in the proposed pathway. It seems to be a promising method to stabilized reactive lignin fragments formed under basic conditions. I recommend publication in Nature Communications after attention is paid to the points made above.

Reviewer #2 (Remarks to the Author):

Development of alternative route to N-containing aromatic heterocyclic compounds from renewable resource is of great importance to achieving carbon-neutrality and alleviating the reliance on fossil resource. In this manuscript, Zhang and co-workers reports a unique transformation which enables the lignin β -O-4 model compounds directly transform into pyrimidines via a one-pot multi-component cascade reaction. The experiments have been carefully designed and the reaction mechanism has been well studies. Moreover, the application feasibility in pharmaceutical synthesis has been demonstrated.

This study is related to the hot field of lignin upgrading, the idea of converting lignin model compounds to value-added pyrimidines heterocyclic aromatic compounds is of highly novelty and importance. To my knowledge, this is the first example of the in-situ construction of heterocyclic aromatic compounds from lignin models and should be a major breakthrough in lignin conversion. The article is compelling and timely, and I believe it should be suitable for publication in Nature Communications after minor modifications.

1. In the proposed reaction pathway, β -O-4 model compound 1a was expected to be able to undergo a NaOH-induced deprotonation of C α -H bond before the cleavage of the C-O bond. However, there is another possible way of direct cutting the C-O bond before the deprotonation of C α -H bond. A reaction using 1-phenylethane-1,2-diol in replace of β -O-4 compound should be done to prove/disprove the argument.

2. I noticed that excessive amount of NaOH was used in the reaction. As the whole transformation involves relatively complex sequential cleavage of C-O bonds, alcohol dehydrogenation, aldol condensation, and dehydrogenative aromatization, which step needs excessive amount of NaOH? Cleavage of C-O bonds (the first step), or the cross-aldol condensation step? More comparative experiments are suggested to figure out the role of NaOH. Meanwhile, is it possible to replace NaOH by solid base catalyst?

3. My other concern is that all these reactions are very dilute and removing the solvent would be very expensive. Does this reaction work when the authors do the reaction in a more concentrated feedstock (like 10 wt% feed)? I recommend that the authors give this data with one of the substrates and added some discussion about this issue before publication.

4. For some low conversion substrates such as the lignin model with γ -OH group, higher temperature might help increasing the conversion.

Reviewer #3 (Remarks to the Author):

This work presented results on production of pyrimidines from a lignin segment. The work is good one but writing is very poor. I don't recommend its publication because of following concerns.

The novelty of the work is also not clear and not up to the standards of the Nature Communication journal.

This work and process is similar to fraction of lignocellulosic biomass in to lignin, cellulose and hemicellulose using either acid or alkali based thermal treatment. In such methods, often several VAC produced and which are just need to be segregated from the product mixture; and that is what followed in this work as I understand because of several details of experimental steps are missing in the manuscript.

Several such works have already been carried out and published which is evident from the references 10-24 of this manuscript. This makes the present work a routine incremental one.

Also lot of similarity with authors' previous work [24].

Details of DFT are completely missing, e.g., details of basis set, functional, coordinates, etc.

DFT protocol has to be first established by validating with certain experimental results existing in the literature before adopting it for new mechanisms

Details of solvent and SMD parameters should be presented for reliability check.

The ω B97X-D/6-311+g(d,p) level is not sufficient for handling big molecules such as the lignin segment considered in this work.

Response to Reviewers (NCOMMS-21-37866-T)

Transition-metal-free synthesis of pyrimidines from lignin β -O-4 segments via a one-pot multi-component reaction

Bo Zhang, Tenglong Guo, Zhewei Li, Fritz E. Kühn, Ming Lei, Zongbao K. Zhao, Jianliang Xiao, Jian Zhang,
Dezhu Xu, Tao Zhang, Changzhi Li

Dear reviewers,

Thank you for the time and effort you have spent in reviewing our manuscript. We particularly appreciate the insightful and constructive comments. We carefully revised and improved our manuscript following the comments and suggestions. The detailed replies and corrections are listed below point by point. All the changes are highlighted in blue in the revised manuscript.

Reviewer #1 recommendation: Publish after minor revisions noted.

Remarks to the Author:

In the paper "Transition-metal-free synthesis of pyrimidines from lignin β -O-4 segments via a one-pot multi-component reaction" an approach to address the sourcing, synthesis, and the mechanism behind the formation of pyrimidines containing bio-based fine chemicals is presented. This paper is important for bridging the gap between the fundamental problems of fossils dependence and implement on of bio-based alternatives into the market. Starting from model compounds mimicking lignin main unit, the authors performed elegant cascade of reactions to generate pyrimidine derivatives in fair yields (based on lignin part) and reasonable selectivity. The mechanism of the reaction is partially studied experimentally and thoroughly supported by the computational analysis. The synthesis of a meridianin derivative performed to highlight applicability of the method. This is a challenging topic and it is well addressed by the authors.

I have divided my comments into major A# and minor B#:

A1. *The yield of all compounds is calculated based on a benzamidine derivatives. It would be useful to point out what is the wt% incorporation of the lignin derived part in the final products since both they*

are used in excess and stated conversion is 100%. The authors should include that information in the main text and also for the best presented case and meridianin derivative. For example, for the derivative **4c**, that is the most relevant one, the yield would be 19% (ca 14.7 wt% of the initial lignin model is incorporated).

Author reply: We appreciate this reviewer's positive comments on our work. We apologize for the blurry description of the yield of the major products. As benzamidine derivatives **2** are obtained from fossil resource while **1** and **3** are renewable lignin-derived compounds, the latter two substrates are used in excessive in the reaction, therefore, the molar yield of pyrimidine product **4** was calculated based on benzamidine derivative **2**. For phenolic product **5**, it is completely produced from cleavage of **1**, thus the molar yield of **5** was calculated based on lignin model compound **1**. We have added the above information as table caption in Tables 1 and 2 to make the results clearer.

According to the suggestion, the mass yields of the pyrimidine products based on the lignin model compounds were discussed in the revised manuscript on Pages 7-8, and listed in Tables S4-S5 in Supporting Information. For the corresponding description, please find on Page 6 highlighted in blue, to read, "After screening of the reaction parameters including solvents, base loading, reaction temperature and time (Tables S1-S2 and Figure S1), the optimized condition was identified, leading to 95 mol% gas chromatograph (GC) yield (93 mol% isolated yield) of **4a** based on the amount of **2a**, which was accounted for 59 wt% **4a** yield based on lignin model compound **1a**, along with 99 mol% GC yield (95 mol% isolated yield) of guaiacol **5a** (Table 1, entry 1)." And Page 8-9, to read "The mass yields of all pyrimidines **4** were in the range of 41 wt%-59 wt% based on the amounts of lignin model compounds **1** (Table S4). Specifically, the highly substituted β -O-4 model compound **1g** containing γ -OH functionality was also tolerated, smoothly providing the target product **4c** in 38% yield (21 wt% yield based on **1g**) (Table 2, entry 7) although it has a more complicated structure and contains higher steric hindrance compared to **1a-1f**."

As suggested, the mass yield of meridianin derivative based on the excessive lignin model compound **1d** was also added into the revised manuscript on Page 19, highlighted in blue. The detailed calculation equation was added into Supporting Information on Pages S5, the first paragraph, to read "Lignin model compound **1d** (110 mg, 0.4 mmol), guanidine hydrochloride **2g** (19 mg, 0.2 mmol), (1-benzyl-1H-indol-3-yl)methanol **3g** (95 mg, 0.4 mmol), NaOH (64 mg, 1.6 mmol), *t*-AmOH (4 mL) were placed in a pressure tube. The mixture was sealed and heated to 110 °C for 20 h. After

reaction, the solution was cooled to room temperature, and ethyl acetate (3 mL) was added into the mixture. The solvent was evaporated under reduced pressure, and the crude products were purified by column chromatography using petroleum ether/ethyl acetate (2:1) to obtain the desired products **10** (56 mg, 69% yield, see Page S33 in Supporting Information for NMR spectra of **10**).

A mixture of **10** (41 mg, 0.1 mmol) and *t*-BuOK (80 mg, 0.7 mmol) in 1 mL DMSO was stirred at room temperature under atmospheric oxygen atmosphere for 8 h. Upon completion by TLC monitoring, the reaction was quenched with saturated aqueous NH₄Cl (10 mL), and extracted with EtOAc (3×10 mL). The combined organic phase was dried over anhydrous Na₂SO₄ and concentrated under reduced pressure, then was isolated by flash silica gel column chromatography (petroleum ether (30-60 °C)/diethyl ether = 2:1, v/v) to afford **11** as a white solid (22 mg, 70% yield). The overall yield of **11** based on **1d** is 27 wt% (calculation formular: 22 mg*(56 mg/41 mg)/110 mg*100%). Compound **11** was dissolved in *d*₆-DMSO and transferred into NMR tubes for NMR characterization; NMR spectra of compound **11** can be seen on Page S34. The purity is >95% according to NMR results.”

A2. *Among model compounds there are no examples of the terminal β-O-4 groups (non-methylated hydroxy group in the para-position to the main chain/backbond). That is an important part that should be added and such model compound need to be tested. Terminal and internal groups are known to react in a very different way.*

Author reply: We totally agree with this comment that the terminal and internal groups react in a very different way. As suggested, the experiment using the β-O-4 model compound with -OH group in the *para*-position (4-(1-hydroxy-2-phenoxyethyl)phenol) as a substrate was carried out under the condition as that in Entry 1, Table 1. The results showed that this reaction failed to occur. This is mainly because that the OH group of β-O-4 model compound easily reacts with NaOH to form sodium 4-(1-hydroxy-2-phenoxyethyl)phenolate as a salt under base condition, which is hard to dissolve in *tert*-amyl alcohol, resulting in troublesome in mass transfer so that the cleavage of C-O bond in β-O-4 model compound (the first step) cannot initiate.

A3. *It is stated in the article that biomass derived lignin is foreseen. Therefore, a model β-O-4 polymer need to be tested. That would demonstrate the solvent system and reactions conditions are compatible with high molecular weight compounds.*

Author reply: As suggested, we have synthesized the β -O-4 polymer ($M_n=2341$, $M_w=6476$) according to reported method (*J. Am. Chem. Soc.* **2010**, *132*, 12555; *ChemSusChem.* **2015**, *8*, 2187). The β -O-4 polymer was directly used as a substrate mediated by NaOH under the optimized condition (Table 1, Entry 1). Unfortunately, this reaction was hard to occur due to the same reason as that of A2 question, i.e. the reactive hydrogen of OH group at the para-position of β -O-4 polymer easily reacted with Na ion to produce the ONa salt. Despite we did not realize the one-pot conversion of the β -O-4 polymer to pyrimidine, an alternative process composed of a hydroxyl protection step and our key reaction was developed, which achieved 66 wt% yield of pyrimidine derivative **14** based on β -O-4 polymer (Scheme S1), showing the developed reaction could be employed in the conversion of high molecular weight compounds. We have added the corresponding results and discussion in the revised manuscript on Page 11. To read “To further determine the compatibility of the catalytic system, a β -O-4 polymer that mimicking natural lignin was prepared and was used as a substrate for the synthesis of pyrimidine derivative. Though the one-pot, direct conversion of the β -O-4 polymer to pyrimidine was not achieved by reason that the reactive OH group at the *para*-position of β -O-4 polymer easily reacted with Na ion to produce sodium salt under base environment, an alternative three-step process composed of our key reaction was developed, which achieved 66% overall yield of pyrimidine derivative **4p** based on β -O-4 polymer (Scheme S1): First, binuclear rhodium complex-catalyzed mild depolymerization of β -O-4 polymer led to 4-hydroxyacetophenone (compound **I**) in 75% isolated yield, then 4-hydroxyacetophenone reacted with benzyl bromide in the presence of K_2CO_3 to afford 1-(4-(benzyloxy)phenyl)ethan-1-one (compound **II**) in 95% isolated yield, which subsequently reacted with benzamidine hydrochloride and benzyl alcohol, successful providing pyrimidine derivative **4p** in 92% isolated yield.

Scheme S1. Synthesis of pyrimidine derivative **4p** from β -O-4 polymer.

The detailed procedure was added into Supporting Information on Pages S5-S6 . To read:

2.3 Typical procedure of β -O-4 polymer conversion to 4-(4-(benzyloxy)phenyl)-2,6-diphenylpyrimidine (compound **4p**)

β -O-4 polymer was prepared according to literature.^{1,2} 4-Hydroxyacetophenone (**I**) was obtained by depolymerization of β -O-4 polymer over binuclear rhodium complex according to our previous work.³ In detail, β -O-4 polymer (100 mg), NaOH (32 mg) and the binuclear rhodium catalyst (8 mg, 1 mol%), and H₂O (2 mL) were added into a pressure tube under argon atmosphere. The reaction was performed at 110 °C for 18 h. After cooling to room temperature, hydrochloric acid (1 M) was used to acidify the aqueous solution to PH = 1, which was then extracted with ethyl acetate for three times. The organic layer was combined, washed with brine and dried over anhydride MgSO₄. The solvent was evaporated under reduced pressure. The residue was purified by column chromatography using petroleum ether/ethyl acetate (5:1) to obtain the desired products **I** (75 mg, 75% yield).

Synthesis of compound **II** was based on the previous paper.⁴ A mixture of **I** (75 mg, 0.55 mmol), benzylbromide (112 mg, 0.66 mmol), and K₂CO₃ (182 mg, 1.32 mmol) in DMF (5 mL) was stirred at room temperature for 72 h. After reaction, water (5 mL) was added into the solution. After filtration, the filtrate was extracted with diethyl ether and the organic extracts were concentrated and purified by column chromatography using petroleum ether/ethyl acetate (5:1) to obtain the desired products **II** (118 mg, 95% isolated yield).

Compound **II** (118 mg, 0.52 mmol), benzamidine hydrochloride **2a** (41 mg, 0.26 mmol), phenylmethanol **3a** (56 mg, 0.52 mmol), NaOH (84 mg, 2.1 mmol), *t*-AmOH (4 mL) were placed in a pressure tube (35 mL). The mixture was sealed and heated to 110 °C for 20 h. After reaction, the solution was cooled to room temperature, and ethyl acetate (3 mL) was added into the mixture. The solvent was evaporated under reduced pressure, and the crude products were purified by column chromatography using petroleum ether/ethyl acetate (2:1) to obtain the desired products **4p** (99 mg, 92% isolated yield; 66% yield based on β -O-4 polymer, see Page S35 for NMR spectra of **4p**).

4-(4-(benzyloxy)phenyl)-2,6-diphenylpyrimidine (compound 4p): 99 mg, isolated yield 92%.

White solid, m.p.: 104-106 °C. ^1H NMR (400 MHz, CDCl_3) δ 8.75 (m, 2 H, aromatic CH), 8.33-8.25 (m, 4 H, aromatic CH), 7.94 (s, 1 H, aromatic CH), 7.61-7.53 (m, 6 H, aromatic CH), 7.49 (d, $J = 7.2$ Hz, 2 H, aromatic CH), 7.44 (t, $J = 7.3$ Hz, 2 H, aromatic CH), 7.38 (d, $J = 7.1$ Hz, 1 H, aromatic CH), 7.14 (d, $J = 8.8$ Hz, 2 H, aromatic CH), 5.16 (s, 2 H, PhCH_2). $^{13}\text{C}\{^1\text{H}\}$ (100 MHz, CDCl_3) δ 164.5, 164.4, 164.2, 161.1, 138.4, 137.8, 136.6 and 130.2 (Cq each), 130.8, 130.7, 129.0, 128.9, 128.8, 128.5, 128.3, 127.6, 127.3, 115.2 and 109.5 (CH), 70.2 (PhCH_2). HRMS Calcd for $\text{C}_{29}\text{H}_{23}\text{N}_2\text{O}[\text{M}+\text{H}]^+$: 415.1810; Found: 415.1814.

A4. There is no supplementary information (procedure, etc.) for the Reactions presented in Figure 2 in the SI. Please add it.

Author reply: We thank this reviewer for pointing out the carelessness. We have added the procedure of Figure 2 in Supporting Information on Page S4 highlighted in blue. To read:

2.1 Typical procedures of the reactions in Figure 2.

Typical procedure of Reaction (1) in Figure 2: lignin model compound **1a** (0.4 mmol) NaOH (1.6 mmol), internal standard mesitylene (8 mg), and *t*-AmOH (4 mL) were added in a pressure tube (35 mL). The mixture was sealed and heated to 110 °C for 1 h. After reaction, the solution was cooled to room temperature, and ethyl acetate (6 mL) was added into the mixture. Then hydrochloric acid (2M) was used to acidify the aqueous solution to pH = 1. The organic phase was analyzed by GC-FID to determine the yield of guaiacol **5a** using mesitylene as an internal standard. Then the solvent was evaporated under reduced pressure, and the crude products were purified by column chromatography using petroleum ether/ethyl acetate (9:1) to obtain 71% yield of acetophenone **6**.

Typical procedure of Reactions (2-4, and 6) in Figure 2: lignin model compound **1a** (0.2 mmol) or acetophenone **6** (0.2 mmol), the primary alcohol **3a** or benzaldehyde **7** (0.2 mmol), benzamidine hydrochloride **2a** (0.1 mmol), NaOH (0.8 mmol), internal standard mesitylene (4 mg), and *t*-AmOH (2

mL) were added in a pressure tube (35 mL). The mixture was sealed and heated to 110 °C for 20 h under air. After reaction, the solution was cooled to room temperature, and ethyl acetate (3 mL) was added into the mixture. Then hydrochloric acid (2M) was used to acidify the aqueous solution to pH = 1. The organic phase was analyzed by GC-FID to determine the yields of guaiacol **5a** and 2,4,6-triphenylpyrimidine **4a** using mesitylene as an internal standard.

Typical procedure of Reaction (5) in Figure 2: (*E*)-chalcone **8** (0.2 mmol) and benzamidine hydrochloride **2a** (0.1 mmol), NaOH (0.8 mmol), internal standard mesitylene (4 mg), and *t*-AmOH (2 mL) were placed in a pressure tube (35 mL). The mixture was sealed and heated to 110 °C for 20 h. After reaction, the solution was cooled to room temperature, and ethyl acetate (3 mL) was added into the mixture. Then hydrochloric acid (2M) was used to acidify the aqueous solution to pH = 1. The organic phase was analyzed by GC-FID using mesitylene as an internal standard to afford 59% yield of 2,4,6-triphenylpyrimidine **4a**.”

A5. *There is no supplementary information for the synthesis presented in Figure 4. Please add it (amounts, purification etc.).*

Author reply: As suggested we have added the detailed procedure of the reaction in Figure 4 in Supporting Information on Page S5 highlighted in blue. To read:

2.2 Typical procedure for the synthesis of meridianin derivative (compound 11)

Lignin model compound **1d** (110 mg, 0.4 mmol), guanidine hydrochloride **2g** (19 mg, 0.2 mmol), (1-benzyl-1H-indol-3-yl)methanol **3g** (95 mg, 0.4 mmol), NaOH (64 mg, 1.6 mmol), *t*-AmOH (4 mL) were placed in a pressure tube (35 mL). The mixture was sealed and heated to 110 °C for 20 h. After reaction, the solution was cooled to room temperature, and ethyl acetate (6 mL) was added into the mixture. The solvent was evaporated under reduced pressure, and the crude products were purified by column chromatography using petroleum ether/ethyl acetate (2:1) to obtain the desired product **10** (56 mg, 69% yield, see Page S33 for ¹H and ¹³C NMR spectra).

A mixture of **10** (41 mg, 0.1 mmol) and *t*-BuOK (80 mg, 0.7 mmol) in 1 mL DMSO was stirred at room temperature under atmospheric oxygen atmosphere for 8 h. Upon completion by TLC monitoring, the reaction was quenched with saturated aqueous NH₄Cl (10 mL), and extracted with EtOAc (3×10 mL). The combined organic phase was dried over anhydrous Na₂SO₄ and concentrated under reduced pressure. Isolation by flash silica gel column chromatography (petroleum ether (30-60

°C)/diethyl ether = 2:1, v/v) afforded **11** as a white solid (22 mg, 70% yield). The overall yield of **11** based on **1d** is 27 wt% (calculation formular: 22 mg*(56 mg/41 mg)/110 mg*100%). Compound **11** was dissolved in *d*₆-DMSO and transferred into NMR tube for NMR characterization. The purity is > 95% according to NMR results (Page S34 for ¹H and ¹³C NMR spectra).

A6. *How the amount of compound 3a was justified? Is there any optimization available?*

Author reply: The lower amount of compound **3a** lead to the lower yield of the target product **4a**. For example, the mole ratio of **3a** : **2a** = 2:1, 1.5:1 and 1:1 gave **4a** yield of 95%, 75% and 50%, respectively. Therefore, the ratio of 2:1 for **3a** : **2a** is the preferred condition. As suggested, we have added these results in Table S3 in Supporting Information on Page S8.

A7. *Authors proposed oxygen as the oxidizing agent in the reaction supported by calculations. Please perform reactions Table 1, entry 1 and Figure 2, line a) under inert atmosphere. That will both strength the statement and also exclude the possibility that fragments formed from 1a are possible oxidants.*

Author reply: Thank you for this valuable suggestion. As suggested, we have performed the reactions (Entry 1 in Table 1 and Reaction 2 in Figure 2) under inert atmosphere. The results showed that 13% and 68% yields of product **4a** were obtained, respectively, indicating that acetophenone itself acted as another hydrogen-acceptor. DFT calculation further support the above fact (Figure R1-2, Figure 3). Benzyl alcohol **3a** reacts with sodium hydroxide via **TSC1-2** to generate **C2** with an energy barrier of 0.2 kcal/mol (Figure R1). In path C using O₂ as the hydrogen acceptor, **C2** interacts with O₂ to achieve **C3**, followed by the dehydrogenation of **C3**. The energy barrier for the dehydrogenation step via **TSC3-4** is 10.8 kcal/mol. And in path C-AP using acetophenone **6** as the hydrogen acceptor, the energy barrier of the dehydrogenation via **TSC3-4-AP** is 17.5 kcal/mol. This is consistent with experimental observation in Figure 2. Similarly, the dehydrogenations of compound **9** using O₂ in air (path D) and **6** (path D-AP) as hydrogen acceptors were also investigated using DFT method (see Figure R2). **6** reacts with sodium hydroxide via **TSD1-2** to generate **D2**. The energy barrier of path D from **D3** to **D4** is 13.2 kcal/mol, which is lower than that of path D-AP from **D3-AP** to **D4-AP** (13.0 kcal/mol). In addition, the by-product NaOOH in path C and path D could oxidize **3a** or **9** in path C-O and release two NaOH molecules (Figure S2).

Figure R1. The Gibbs free energy profiles for the dehydrogenation of benzyl alcohol **3a** using O_2 in air (path C) and **6** as hydrogen acceptors (path C-AP) (unit: kcal/mol).

Figure R2. The Gibbs free energy profiles for the dehydrogenation of 2,4,6-triphenyl-1,2-dihydropyrimidine **9** using O_2 in air (path D) and **6** as hydrogen acceptors (path D-AP) (unit: kcal/mol).

We have added the control experiment under inert atmosphere as Reaction 6 in Fig. 2. And the Gibbs free energy profiles for the dehydrogenation using acetophenone as a hydrogen acceptor were added in the Fig. 3d and 3e. Also the corresponding discuss was added in the revised manuscript on Pages 16-18, highlighted in blue. To read: “Therefore, a control experiment under argon atmosphere was

conducted, which resulted in a much lower **4a** yield (68%, Reaction 6 in Fig. 2) compared to that in air (99%, Reaction 2 in Fig. 2), illustrating that oxygen in air acted as a hydrogen acceptor in the reaction. It also should be noted that 1-phenylethanol was detected, indicating that intermediate **6** (acetophenone) acted as another hydrogen acceptor. With the aim to verify this speculation and better understand the dehydrogenation mechanism, further DFT calculation was performed. The results, summarized in Fig. 3d, show that benzyl alcohol **3a** reacts with sodium hydroxide via **TSC1-2** to generate **C2** with an energy barrier of 0.2 kcal/mol. In path **C** using O₂ as the hydrogen acceptor, **C2** interacts with O₂ to achieve **C3**, followed by the dehydrogenation of **C3**. The energy barrier for the dehydrogenation step via **TSC3-4** is 10.8 kcal/mol. And in path **C-AP** using acetophenone **6** as a hydrogen acceptor, the energy barrier of the dehydrogenation via **TSC3-4-AP** is 17.5 kcal/mol. This is consistent with experimental observation in Reaction 2 Fig. 2. In addition, the acceptor-free dehydrogenation pathway was also calculated under the same condition, giving two possibilities. As shown in Figure 3d and Figure S3, for acceptor-free dehydrogenation, the energy barrier via **TSC2-3-B1** would be 34.4 kcal/mol, which is higher than in path **C**. Moreover, the experiment results further suggest that no hydrogen was detected during the reaction. Combining the DFT calculation and experimental results, it can be concluded that acceptor-free dehydrogenation **did** not occur for the dehydrogenation of benzyl alcohol.

Similarly, the dehydrogenation of intermediate **9** using both O₂ in air (path D) and **6** (path D-AP) as hydrogen acceptors was also studied. For comparison an acceptor-free dehydrogenation pathway (path D-B) was also investigated by DFT calculation. As shown in Fig. 3e, **9** first reacts with sodium hydroxide via **TSD1-2** to generate **D2**. Then **D2** follows path **D-B** to generate pyrimidine **4a** in the absence of a hydrogen acceptor with an energy barrier of 30.1 kcal/mol. Alternatively, **D2** goes

through **D3** and **D4** to form **4a** in the presence of air with a much lower energy barrier of 3.2 kcal/mol, which is lower than that of path **D-AP** from **D3-AP** and **D4-AP** (13.0 kcal/mol), and H₂ is not detected during the reaction. Apparently, the pathway from **9** to **4a** is also not an acceptor-free dehydrogenation reaction. The DFT results summarized in Fig. 3d and Fig. 3e illustrate that O₂ in air acts as a hydrogen acceptor for the dehydrogenation of benzyl alcohol **3a** to **7** and **9** to **4a**. In addition, the by-product NaOOH in path C and path D could oxidize **3a** or **9** to release two NaOH molecules (Figure S4).”

B1. Page 6. Part of the text is devoted to the justification of the guaiacol formation and its usefulness as a chemical. I would recommend to remove that part since the described process won't result in this product when a relevant starting material will be used.

Author reply: We have removed it according to this suggestion.

B2. The part where the mechanism is studied is lacking of references. Most of the reactions are known and there is no literature cited. Please check for all the steps discussed in this part.

Author reply: As suggested we have added the corresponding references.

B3. Table S1. Please remove duplicates.

Author reply: As suggested we have removed the duplicates in Table S1.

B4. In the descriptions of the spectral data in the SI there is no reference to the known compounds' spectral data. Was the data compared and verified that it is matching the literature? Please add references and verify that the obtained spectral data is in accordance with the previously reported.

Author reply: We have checked that the spectral data in the SI is in accordance with the previously reported. We have added the corresponding references in the SI as suggested.

Summary comments: Overall, the authors have presented a novel system that potentially can convert lignin from wood to fine chemicals containing pyrimidines. There is considerable novelty and value in the proposed pathway. It seems to be a promising method to stabilize reactive lignin fragments

formed under basic conditions. I recommend publication in Nature Communications after attention is paid to the points made above.

Author reply: We thank this reviewer again for this positive comment and above precious suggestions on our work.

Reviewer #2 recommendation: Publish after minor revisions noted.

Remarks to the Author:

Development of alternative route to N-containing aromatic heterocyclic compounds from renewable resource is of great importance to achieving carbon-neutrality and alleviating the reliance on fossil resource. In this manuscript, Zhang and co-workers reports a unique transformation which enables the lignin β -O-4 model compounds directly transform into pyrimidines via a one-pot multi-component cascade reaction. The experiments have been carefully designed and the reaction mechanism has been well studies. Moreover, the application feasibility in pharmaceutical synthesis has been demonstrated.

This study is related to the hot field of lignin upgrading, the idea of converting lignin model compounds to value-added pyrimidines heterocyclic aromatic compounds is of highly novelty and importance. To my knowledge, this is the first example of the in-situ construction of heterocyclic aromatic compounds from lignin models and should be a major breakthrough in lignin conversion. The article is compelling and timely, and I believe it should be suitable for publication in Nature Communications after minor modifications.

Author reply: We deeply appreciate this reviewer for the comments.

Q1. *In the proposed reaction pathway, β -O-4 model compound 1a was expected to be able to undergo a NaOH-induced deprotonation of $C\alpha$ -H bond before the cleavage of the C-O bond. However, there is another possible way of direct cutting the C-O bond before the deprotonation of $C\alpha$ -H bond. A reaction using 1-phenylethane-1,2-diol in replace of β -O-4 compound should be done to prove/disprove the argument.*

Author reply: As suggested the experiment using 1-phenylethane-1,2-diol as a substrate mediated by NaOH was performed under the same condition as that in entry 1 Table 1. The results showed that no reaction occurred, illustrating that NaOH induced deprotonation of $C\alpha$ -H bond before the cleavage of the C-O bond. Accordingly, this experiment excludes the possibility of direct cutting the C-O bond.

Q2. I noticed that excessive amount of NaOH was used in the reaction. As the whole transformation involves relatively complex sequential cleavage of C-O bonds, alcohol dehydrogenation, aldol condensation, and dehydrogenative aromatization, which step needs excessive amount of NaOH? Cleavage of C-O bonds (the first step), or the cross-aldol condensation step? More comparative experiments are suggested to figure out the role of NaOH. Meanwhile, is it possible to replace NaOH by solid base catalyst?

Author reply: To answer this question, we have reduced base amount in the first step (C-O bond cleavage, Reaction 1, Fig. 2). The results showed that the reaction is hard to occur, indicating that concentrated base plays a crucial role for the cleavage of C-O bonds. It is also known that cross-aldol condensation easily occurs under base condition (Dumesic, et al. *Science*, 2005, 308, 1446), and dehydrogenation of alcohol could also occur under basic environment (Rahul, et al. *Eur. J. Org. Chem.* 2020, 3081). According to the above results and literatures, we propose that NaOH plays a multifunctional role for the *sequential cleavage of C-O bonds, alcohol dehydrogenation, aldol condensation, and dehydrogenative aromatization*. Specifically, excessive base is crucial for the C-O bond cleavage step in our reaction system.

According to the comments, we have also tested several solid bases such as CaO, MgO, and $\text{Mg}_6\text{Al}_2(\text{CO}_3)(\text{OH})_{16}$. The results showed that the reaction failed to occur, illustrating that these solid bases could not promote this complex cascade transformation in our systems. We have added these results in Supporting Information, Table S1 on Page 7, highlighted in blue.

Q3. My other concern is that all these reactions are very dilute and removing the solvent would be very expensive. Does this reaction work when the authors do the reaction in a more concentrated feedstock (like 10 wt% feed)? I recommend that the authors give this data with one of the substrates and added some discussion about this issue before publication.

Author reply: We apologize we did not discuss the recycle of the solvent in the first submission. It should be noted that the solvent *tert*-amyl alcohol can be easily distilled from the reaction mixture for recycling because the boiling point of *tert*-amyl alcohol (101.8 °C) is much lower than that of the reactants (for example, **1a**: 398 °C; **2a**: 208 °C; **3a**: 205 °C) and the products (**4a**: 330 °C; **5a**: 205 °C). Hence, it is actually an easy process for recycling of the solvent in our system. We have added this point in the revised manuscript (Page 6, highlighted in blue), to read “After reaction, the solvent

tert-amyl alcohol can be easily distilled from the reaction mixture for recycling because the boiling point of *tert*-amyl alcohol (101.8 °C) is much lower than that of all the reactants (for example, **1a**: 398 °C; **2a**: 208 °C; **3a**: 205 °C) and the products (**4a**: 330 °C; **5a**: 205 °C)."

As suggested, the loading of **1a** was increased to 10.8 wt% concentration, correspondingly, the concentrations of **2a** and **3a** were also increased to four folds of their original amounts. Despite this multicomponent reaction system became viscous due to the high concentrations of all three substrates, **4a** and **5a** were still obtained in yields of 10% and 14%, respectively, indicating that the reaction also occurred with highly concentrated substrates. However, on account that it is a multi-component transformation with different functional groups on the substrates, the reaction with concentrated feedstocks would induce severe side reactions on different substrates and intermediates, hence, the yield of the targeted products decreased substantially. We have added the new conversion results in Table S1 entry 16 and the corresponding discussion into the revised manuscript in the first paragraph on Page 8, highlighted in blue, to read "Moreover, the reaction can be proceeded at higher substrate concentration (Entry 16, Table S1) but the yields of the targeted products decreased due to severer side reactions."

Q4. *For some low conversion substrates such as the lignin model with γ -OH group, higher temperature might help increasing the conversion.*

Author reply: As suggested we have increased the reaction temperature to 140 °C during the conversion of lignin model with γ -OH group. Unfortunately, the yield of the target product **4c** was not obviously improved probably due that the elevated temperature might induce annoying side reactions as we have replied in **Q3** of this reviewer.

Reviewer #3 (Remarks to the Author):

1. This work presented results on production of pyrimidines from a lignin segment. The work is good one but writing is very poor. I don't recommend its publication because of following concerns. The novelty of the work is also not clear and not up to the standards of the Nature Communication journal. This work and process is similar to fraction of lignocellulosic biomass in to lignin, cellulose and hemicellulose using either acid or alkali based thermal treatment. In such methods, often several VAC

produced and which are just need to be segregated from the product mixture; and that is what followed in this work as I understand because of several details of experimental steps are missing in the manuscript.

Author reply: We thank this reviewer for the comments. After reading the words from this reviewer, we try our best to improve the manuscript according to the comments.

The reviewer commented that “The novelty of the work is not clear, this work and process is similar to fraction of lignocellulosic biomass in to lignin, cellulose and hemicellulose using either acid or alkali based thermal treatment...” ***We do not think, however, the major merits and originality of our work were fully identified or recognized by this reviewer. Indeed, it is a known case that neither acid nor alkali based thermal treatment of lignocellulosic biomass can achieve the selective production of heterocycle aromatic compounds, particularly pyrimidines,*** the crucial building blocks in pharmaceutical synthesis. We are confident that our work has clear originality based on the following reasons: It is well known that depolymerization of lignin only provides C, H, O-containing products. So far, dominant research efforts have been dedicated to controllable cleavage of the C-O and C-C bonds in lignin to obtain low molecular weight aromatics. Despite several literatures realize the conversion of lignin model compounds to N-containing aromatic products, ***state-of-the-art N-participated lignin conversion primarily focuses on amination of lignin-derived monomers or modified dimer model compounds to afford ARYLAMINE products in the presence of external oxidant or reductant species catalyzed by transition-metals. Synthesis of N-heterocyclic PYRIMIDINES from lignin-derived feedstock still remains a blank,*** the major challenges are the structural complexity of lignin, the incompatible catalysis for C-O cleavage and C-N formation, as well as the in-situ N-heterocyclic ring construction.

In this work, we report the first example of PYRIMIDINES synthesis from lignin 6-O-4 model compounds, the most abundant segments in lignin. Such a transformation undergoes a highly-coupled multi-component cascade reaction in a one-pot fashion, and NaOH as a sole catalyst plays a multifunctional role, rendering the process highly efficient and outstanding selectivity towards the formation of pyrimidines. ***This new strategy features transition-metal free catalysis, no need of external oxidant/reductant, in-situ nitrogen-heterocyclic ring construction, highly coupled multi-step transformation, and sustainable universal approach, thus provides an unprecedented opportunity for N-heterocyclic aromatic compounds production from renewable biomass.*** With this

protocol, a wide range of functionalized pyrimidines including an important marine alkaloid meridianin derivative have been synthesized, emphasizing the application potential in pharmaceutical synthesis. ***From above cases, we believe this methodology paves a new way for the fabrication of heterocyclic aromatic compounds from lignin, thus providing a potentially petroleum independent solution to value-added pharmaceutical molecules.***

The originality and importance of this work have also been approved by the other two reviewers. Reviewer 1 commented that “This paper is important for bridging the gap between the fundamental problems of fossils dependence and implement of bio-based alternatives into the market. Starting from model compounds mimicking lignin main unit, *the authors performed elegant cascade of reactions to generate pyrimidine derivatives in fair yields (based on lignin part) and reasonable selectivity. The mechanism of the reaction is partially studied experimentally and thoroughly supported by the computational analysis.* The synthesis of a meridianin derivative performed to highlight applicability of the method...This is a challenging topic and it is well addressed by the authors.”

Reviewer 2 also commented that “Development of alternative route to N-containing aromatic heterocyclic compounds from renewable resource is of great importance to achieving carbon-neutrality and alleviating the reliance on fossil resource...Zhang and co-workers reports a unique transformation...Moreover, the application feasibility in pharmaceutical synthesis has been demonstrated...the idea of converting lignin model compounds to value-added pyrimidines heterocyclic aromatic compounds *is of highly novelty and importance...this is the first example of the in-situ construction of heterocyclic aromatic compounds* from lignin models and *should be a major breakthrough in lignin conversion.*”

Upon the above response to this reviewer’s query, I wish the reviewer could approve of the originality of this work.

For the comment “several details of experimental steps are missing in the manuscript”, this reviewer didn’t clear point out which experimental steps are missing. We have added some experimental details according to A4-A5 of Reviewer #1. We are happy to further modify our experimental details if there is any unclear description.

2. Several such works have already been carried out and published which is evident from the references 10-24 of this manuscript. This makes the present work a routine incremental one. Also, lot of similarity with authors' previous work [24].

Author reply: As has been answered in the former comments, we disagree this reviewer for the comment. As shown in the manuscript and commented by the former two reviewers, **our work is essentially different from previous works (references 10-24) in terms of the first bio-based pyrimidine products, ONE-POT conversion process, transition-metal free catalysis, highly-coupled multi-component cascade reaction mechanism, as well as in-situ construction of nitrogen-heterocyclic ring.** Indeed, the methods reported in references 10-23 concentrated on the treatment of lignin model compounds with transition-metal catalysts for the production of arylamines but not construction of N-heterocyclic compounds. **Particularly for the conversion of lignin β -O-4 model compounds that mimicking lignin main units, hydrogenolysis or oxidation pretreatment is requisite for subsequent amination reaction, and external oxidant or reductant species are essential in such multi-step processes.** To the best of our knowledge, no literature reports the direct conversion of β -O-4 model compounds to **aromatic heterocyclic compounds through in-situ nitrogen-heterocyclic ring construction in the absence of transition metal catalysts and external redox reagents**, due to the extremely complicated reaction path, and the incompatible catalysis for C-O bond cleavage and aromatic nitrogen-heterocyclic ring construction. **Fig. 1 in the manuscript has shown the major originality of this work.**

As for the difference of our own published paper (reference 24), that work focus on the conversion of phenolic β -O-4 model compounds over Pd/C catalyst to benzylamine, but not N-heterocyclic compounds; **the reaction mechanism** (amination of ketone intermediate followed by hydrogenation reaction), **catalyst** (Pd/C), **and the targeted products** (benzylamines) **are all totally different from this manuscript.** We believe our work not only provides a new opportunity for the sustainable synthesis of value-added N-heterocyclic compounds from renewable biomass, but also expands the products pool of biomass conversion to meet future biorefinery demands.

3. Details of DFT are completely missing, e.g., details of basis set, functional, coordinates, etc.

Author reply: Actually, the information of computational method had been described in the method part of the original manuscript. Moreover, the coordinates of stationary points along reaction pathways were also in the Supporting Information part in the original manuscript. Nevertheless, we have revised the section of “Details of DFT calculation” and added more details about calculations in the revised manuscript on Page 20 based on review comments. To read “In this work, ω B97X-D⁴⁷ functional was used for the DFT calculation, which considered the dispersion correction. And the Pople-type triple- ζ split-valence basis sets 6-311+G(d,p) are used for the optimization of all structures. The solvation effect of *tert*-amyl alcohol ($\epsilon=5.78$) was simulated by the SMD⁴⁸ continuum solvent mode. The Cartesian coordinates of all optimized structure are given in the Supporting Information. The calculations were performed using Gaussian 09 program.⁴⁹ All transition states were confirmed with only one imaginary frequency integrated with intrinsic reaction coordinates (IRC)⁵⁰ calculations. All energies discussed above are Gibbs free energies calculated at 298.15 K.”

4. DFT protocol has to be first established by validating with certain experimental results existing in the literature before adopting it for new mechanisms.

Author reply: The calculated results are consistent with the experimental results reported by this work, which prove the existence of intermediates acetophenone and benzaldehyde (entry 4 of Figure 2 of this manuscript). In addition, these results also agree well with previous experimental and theoretical investigation on the mechanisms of transformation of lignin β -O-4 compound under basic condition, such as the mechanism of the cleavage of the β -O-4 compound to form acetophenone investigated by DFT method (*Green Chem.* **2016**, *18*, 1590–1596), and the dehydrogenation of benzyl alcohol to form benzaldehyde reported by experiments (*Green Chem.* **2012**, *14*, 2384-2387). All of these above could validate the reliability of DFT protocol used in this manuscript.

5. Details of solvent and SMD parameters should be presented for reliability check.

Author reply: Thanks a lot for the suggestion. We used *tert*-amyl alcohol as the solvent, which is used in the experiments of this work. In the revised manuscript, the parameters of solvent *tert*-amyl alcohol ($\epsilon=5.78$) have been added in the method part of revised manuscript on Page 20. To read “The solvation effect of *tert*-amyl alcohol ($\epsilon=5.78$) was simulated by the SMD⁴⁸ continuum solvent mode.

The Cartesian coordinates of all optimized structure are given in the Supporting Information. The calculations were performed using Gaussian 09 program.⁴⁹ All transition states were confirmed with only one imaginary frequency integrated with intrinsic reaction coordinates (IRC)⁵⁰ calculations.”

6. The ω B97X-D/6-311+g(d,p) level is not sufficient for handling big molecules such as the lignin segment considered in this work.

Author reply: In this manuscript, ω B97X-D functional was used for the DFT calculation in this work, which considered the dispersion correction. The calculation at a higher level like CCSD method is too expensive and not applicable for this organic system. In order to verify the reliability of the DFT calculations used in this work, the M06-2X functional with the same Pople-type triple- ζ split-valence 6-311+g(d,p) basis set (M06-2X/6-311+g(d,p)) was used to investigate the C-O bond cleavage of the β -O-4 model compound (see Figure R3 below, Figure S5 in the Supporting Information). The calculated results indicate that the calculated free energy profiles at the ω B97X-D/6-311+g(d,p) level and the M06-2X/6-311+g(d,p) level are very similar. And the results of the optimized geometrical structures of stationary points along the reaction pathway of the C-O bond cleavage of the β -O-4 model compound using these two methods are also similar. Meanwhile, in many benchmark articles, this functional has performed well in structural optimization and calculation of reaction energy (*Phys. Chem. Chem. Phys.* **2011**, *13*, 6670-6688; *Molecular Phys.* **2017**, *115*, 2315-2372; *J. Phys. Chem. Lett.* **2020**, *11*, 9957-9964; etc.). Therefore, ω B97X-D/6-311+g(d,p) level is sufficient to deal with the lignin transformation system in this manuscript.

Figure R3. The calculated Gibbs free energy profiles for the C-O bond cleavage of the β -O-4 model

compound **1a** at the ω B97X-D/6-311+g(d,p) level and the M06-2X/6-311+g(d,p) level (unit: kcal/mol).

Reviewer #1 (Remarks to the Author):

All the major points were addressed through revision.

Only one minor note.

Previously proposed method for yield calculations based on wt% was only to demonstrate the degree of lignin incorporation. (see Attachments PDF). Please adjust the yields stated in the article. The yields added after the revision as calculated by the authors should be still in mol% not wt%, and degree of lignin incorporation can be added as well to one of the main examples and meridianin derivative.

Some limitations that were revealed through revision were also addressed. Even though the methods proposed are not optimal especially for a real application the concept and proposed pathway are valuable.

Overall, the authors have presented a novel system that potentially can convert lignin from wood to fine chemicals containing pyrimidines. There is considerable novelty and value in the proposed pathway. It seems to be a promising method to stabilize reactive lignin fragments formed under basic conditions. I recommend publication in Nature Communications.

Reviewer #2 (Remarks to the Author):

The authors have very well addressed all the issues by additional experimental and theoretical studies. In view of the challenge and importance of the lignin valorization, they did a great job in upgrading of lignin models to valuable pyrimidines heterocyclic aromatic compounds, I believe this innovative idea and excellent results will inspire future research efforts. I recommend publish this artwork in Nature Communications as is.

Reviewer #3 (Remarks to the Author):

I appreciate the responses of authors on my earlier suggestions/comments.

REVIEWER 1 Attachment:

Schematic hydrogens are not adjusted

yield on **1g** is 38%

yield on **1d** is 19%

potential lignin incorporation with the stated method is

Part that was used for incorporation $C_{11}H_{15}O_4$ Mw 211

Part that was incorporated $C_{10}H_9O_2$ Mw 161 " CH_4O_2 " was "lost"

Not more than 76.3 wt% of lignin model could be utilized
(even when the yield and selectivity is 100% and mol ratios are 1:1)

Therefore $19 * 0.763 = 14.5$ wt% initial lignin model was utilized

Response to the comments of the reviewers (NCOMMS-21-37866-T)

We thank all reviewers for the insightful and constructive comments. We carefully revised and improved our manuscript following the comments and suggestions. The detailed replies and corrections are listed below point by point. We highlight all the changes in tracked mode in the revised manuscript, and we also provide a revised version with tracked changes accepted.

Reviewer #1 (Remarks to the Author): *All the major points were addressed through revision. Only one minor note. Previously proposed method for yield calculations based on wt% was only to demonstrate the degree of lignin incorporation. (see Attachments PDF). Please adjust the yields stated in the article. The yields added after the revision as calculated by the authors should be still in mol% not wt%, and degree of lignin incorporation can be added as well to one of the main examples and meridianin derivative.*

Author reply: According to the suggestion, the yields stated in the article are adjusted to mol%. The calculated wt% yields in the Supplementary Information have been deleted. And degree of lignin incorporation of **4a** (*one of the main examples*) and meridianin derivative **11** have been calculated according to Attachments PDF as shown below:

For **4a**, the yield on **1a** is 46.5%. Part in **1a** that was used for incorporation: C₈H₁₀O, Mw 122. Part that was incorporated: C₈H₁₀, Mw 101. Hence, no more than 86.9 wt% of lignin model could be utilized. Therefore $46.5 * 0.869 = 40.4$ wt% initial lignin model was utilized.

For the meridianin derivative **11**, the yield on **1d** is 24%. Part in **1d** that was used for incorporation: C₉H₁₂O₂, Mw 152. Part that was incorporated: C₉H₁₂O, Mw 136. Not more than 89.5 wt% of lignin model could be utilized. Therefore $24 * 0.895 = 21.5$ wt% initial lignin model was utilized.

We have added these results in the supplementary information on Page S7. To read:

Part in **1a** that was used for incorporation: $C_8H_{10}O$, M_w 122.

Part that was incorporated: C_8H_{10} , M_w 106. "OH" was "lost".

Not more than 86.9 wt% of lignin model compound could be utilized (even when the yield and selectivity is 100% and mol ratios are 1:1).

Therefore $0.465 * 0.869 = 40.4$ wt% initial lignin model was utilized.

Part in **1d** that was used for incorporation: $C_9H_{12}O_2$, M_w 152.

Part that was incorporated: $C_9H_{12}O$, M_w 136. "OH" was "lost".

Not more than 89.5 wt% of lignin model compound could be utilized (even when the yield and selectivity is 100% and mol ratios are 1:1) Therefore $0.24 * 0.895 = 21.5$ wt% initial lignin model was utilized.

Supplementary Fig. 2. Calculation weight yield of products 4a and 11.